# Multiyear surface waves dataset from the subsurface "DeepLev" Eastern Levantine moored station

Nir Haim[1], Vika Grigorieva[1], Rotem Soffer[1], Boaz Mayzel[1], Timor Katz[2], Ronen Alkalay[2,3], Eli Biton[2], Ayah Lazar[2], Hezi Gildor[4], Ilana Berman-Frank[5], Yishai Weinstein[3], Barak Herut[2,5], and Yaron Toledo[1]

[1]School of Mechanical Engineering, Faculty of Engineering, Tel Aviv University, Tel-Aviv, 6997801, Israel
[2]Israel Oceanographic & Limnological Research, Tel Shikmona, Haifa, 31080, Israel
[3]Department of Geography and Environment, Bar-Ilan University, Ramat-Gan, 52900, Israel
[4]Institute of Earth Sciences, Hebrew University, Jerusalem, 91904, Israel
[5]Leon H. Charney School of Marine Sciences, University of Haifa, Haifa, 3498838, Israel

**Correspondence:** Yaron Toledo (toledo@tauex.tau.ac.il )

**Abstract.**

Processed and analyzed sea surface wave characteristics derived from an up-looking Acoustic Doppler Current Profiler (ADCP) for the period 2016-2022 are presented as a data set available from the public open-access repository of SEA scieNtific Open data Edition (SEANOE) at https://doi.org/10.17882/96904 (Haim et al., 2022). The collected data include full two-dimensional wave fields along with computed bulk parameters, such as wave heights, periods, and directions of propagation. The ADCP was mounted on the submerged Deep Levantine mooring station located 50 km off the Israeli coast to the west of Haifa (bottom depth ∼1470m). It meets the need for accurate and reliable in situ measurements in the Eastern Mediterranean Sea, as the area significantly lacks wave data compare to other Mediterranean sub-basins. The developed long-term timeseries of wave parameters contribute to monitoring and analysis of the region's wave climate, and the quality of wind-wave forecasting models.

## 1 Introduction

In the past decades ocean waves are being observed around the Mediterranean sea. In some cases providing prolonged records (Ntoumas et al., 2022; Vargas-Yáñez et al., 2023; Morucci et al., 2016; Pomaro et al., 2018). More recently, using High-Frequency radars (Lorente et al., 2022). While there are increasing efforts to gather measurements in the sub-basins of the Mediterranean sea (Tintoré et al., 2019) the Levantine basin is still comparably lacking in observations (Toomey et al., 2022). Monitoring ocean waves is crucial for support in making informed decisions related to the development, protection, and management of the marine and coastal environments. Accurate and regular wave measurements are also of great importance in numerous research fields, for example in studying air-sea and wave-current interactions (Wolf and Prandle, 1999), analysing climate changes or investigating the effects of waves dispersion of particles and oil slicks in the water (Fannelop and Waldman, 1972; Sobey and Barker, 1997; Röhrs et al., 2012). Furthermore, the renewable energy sectors seek to harness ocean waves for power generation, and precise wave monitoring is essential for optimizing the design and operation of wave energy converters

(Aderinto and Li, 2018; Lira-Loarca et al., 2021). Including in the Levantine basin, where Zodiatis et al. (2014, 2015) estimated the wave energy potential based on validated wave model.

The acquisition of a long series surface waves data was made possible with the establishment of the Deep Levantine (DeepLev) station that was deployed for the first time on November 2016 about 50 km off-shore Haifa, Israel, at $33°00'N$, $34°30'E$. It was the first of its kind deep ocean moored research station in the ELB conducting measurements across various fields of marine science. Katz et al. (2020) gives a full description of the mooring system and the large number of state-of-the-art measuring instruments it carries. The mooring cable extended from the seabed at depth of approximately 1470 m up to a subsurface buoy (at a nominal depth of $\sim 30$ m) carrying an up-looking Acoustic Doppler Current Profiler (ADCP). In general, instruments for waves measurements are deployed at shallow and intermediate waters (20-40 m depth). A quite understandable practice considering the added complexity and hence increased costs involved in deep sea surveys. Nonetheless, long-term observations at deep waters are valuable for continuous monitoring of sea state. Moreover, avoiding the presence of nearshore bathymetry changes or shore reflections allows for a better accuracy evaluation of wave models and satellite measurements.

In this study, a multi-year open-source dataset of wave spectra and derived wave characteristics (i.e. heights, periods, directions) has been developed from DeepLev station measurements for the period 2016-2022. The paper is organized as follows. Section 2 is dedicated to a general description of the measuring instrument, its operation principles and evaluation of wave information. Section 3 deepens into the collected data, expanding about the processing, issues that emerged and their implications on quality. Conclusive Section 4 finalizes the paper by listing the main results and perspectives of deep sea measurements and wave monitoring in the ELB.

## 2 Methodology

### 2.1 Acoustic Doppler current profiler wave measurements

As it was mentioned above, the DeepLev station is a multi-functional research station, monitoring the sea state and marine environment. Throughout the whole campaign, the Norteks' Signature-500 ADCP was used to measure surface wave parameters thus the derived data are consistent and homogeneous (Figure 1 shows the subsurface buoy and the ADCP mounted on it). The practice of combining of the Nortek's ADCPs and subsurface buoys was found to be successful (Pedersen et al., 2007), though with possible data artifacts due to the buoy's wave induced movement. Compared to Pedersen et al. (2007), in this study the subsurface buoy was deeper therefore expected to be less responsive to surface waves' motion.

The "Signature-500" has three types of sensors: a pressure sensor, four slanted acoustic beams, and a single vertical acoustic beam which gives it an advantage over other types of ADCPs, allowing for several wave field evaluation approaches to be applied. The first method is solely relied on the slanted acoustic beams. The transmitted signals and received Doppler shifted back-scatter (Rowe and Young, 1979; McDaniel and Gorman, 1982) enable to estimate wave characteristics, including the directional wave spectrum, $S^{vel}(f,\theta)$ from the induced orbital velocities near the surface (Bowden and White, 1966). The main limit of the "velocity-based" (hereinafter, VEL) method is its sensitivity to installation depth. In deep installations the

horizontal spacing between the beams increases beyond the solution's validity. Within the DeepLev's settings, the theoretical upper cut-off at 30 m depth is 3.85 sec for directional parameters and 1.15 sec for non-directional.

The second method uses the vertically oriented fifth beam for acoustic surface tracking (AST). The measurement of the surface elevation can be directly represented as a frequency spectrum $S^{ast}(f)$. Here, even short waves which cannot be detected by the slanted beams' array are visible to the AST. Pedersen et al. (2007) offered a way to expand the surface tracking information into directional spectrum, $S^{suv}(f,\theta)$, by combining correlated velocity measurements. This method is known as "SUV" suggesting the combination of surface tracking (S) with horizontal velocities (UV). The name references a third method, the established "PUV" technique (Panicker and Borgman, 1974) which applies similar calculations with pressure observations instead. In this study the depths of installation makes most of the wind-waves frequency range undetectable for the pressure sensor therefore the pressure fluctuations spectrum, $S^{puv}(f)$ is not further discussed though it is included in the dataset as it may be useful to those interested in the low-frequency end of the wind-wave spectrum.

Prior to each deployment the device's operation mode was configured balancing between the expected duration in the sea and available battery capacity. Table 1 summarizes details of the deployments including the configuration of the experiment, its duration, cycle intervals, and sampling frequency. Most of the time, the ADCP was configured to operate with a sampling frequency of 2 Hz, with the exception of the fourth deployment when the sampling frequency was 4 Hz. The cycle intervals are regulated by two different modes of "Signature-500", "Burst" and "Continuous". When set to "Burst mode", the device worked at intervals and collected only 2048 continuous samples withing a cycle (which are equivalent to about 17 min when using 2 Hz). The intervals between cycles were also predetermined and are listed in Table 1. The third and fourth deployments measured in "Continuous" mode without any pauses. For the purpose of consistency, their measurements were analyzed to provide 17 min averages as the rest of the deployments.

## 2.2 Surface waves averages and directional properties extraction

The first stage of data processing was performed by Nortek's "Ocean Contour" Software, which synthesizes the primary binary files into wave information. The simplest type of analysis provided is by directly identifying individual waves in the surface elevation timeseries, $\eta(t)$. Then, wave characteristics are summarized into the maximal measured wave height $H_{max}$ and period $T_{max}$, the mean height $H_{mean}$, mean zero-crossing $T_z$, averages over heights and periods of the highest $1/3$ of the waves, $H_3$ and $T_3$, and over the highest $1/10$ of the waves, $H_{10}$ and $T_{10}$.

Additionally, the timeseries signals are converted using fast Fourier transform into spectral variance density function $S(f)$ (Longuet-Higgins et al., 1963) that indicates how much of the surface wave elevation variance is contained at the specific frequencies $f$. This spectral representation highlights the peak frequency $f_p$, the most energetic frequency inversely related to the peak period $T_p$. Other bulk parameters are calculated through the energy-spectrum's moments (Tucker, 1993), with the moment of order $n$ defined as

$$m_n = \int_0^\infty f^n S(f) df, \tag{1}$$

where $S(f)$ is the directional-averaged density spectrum. The parameters calculated from the spectral moments are the significant wave height $H_{m0} = 4\sqrt{m_0}$, the mean wave period $T_{m02} = \sqrt{\frac{m_0}{m_2}}$, and the energy period $T_{energy} = m_{-1}/m_0$, a weighted mean period based on the spectral density which is useful in estimating wave energy potential. For directional data, the mean wave direction per wave frequency, $\theta_m(f)$, is obtained from the first harmonic Fourier coefficients of the power density spectrum function $S(f,\theta)$ and the corresponding Fourier coefficients $a_n(f)$, $b_n(f)$ as follows

$$\theta_m(f) = \arctan\frac{b_1(f)}{a_1(f)}, \qquad a_n(f) = \frac{1}{S(f)}\int_0^{2\pi} S(f,\theta)\cos n\theta d\theta, \qquad b_n(f) = \frac{1}{S(f)}\int_0^{2\pi} S(f,\theta)\sin n\theta d\theta. \qquad (2)$$

The reported mean wave direction $\theta_m$ is a weighted average of $\theta_m(f)$ in each frequency bin according to the its energy. The peak direction $\theta_p$ is the peak of the spread function constructed employing Fourier coefficients of all available harmonics (n=2) for the peak frequency. Both estimations are expressed here in meteorological conventions, i.e. the specified direction is the direction which the waves are coming from.

The applied methodology provides a complete set of standard wave characterises and allows to compare the results with models, satellites, buoys, and visual wave observations on equal terms.

## 3 Results

The developed dataset presented in this paper includes processed, corrected and analyzed measurements from eight ADCP deployments for the period 2016-2022. In order to save maximum wave information we stored all measurements that passed the original Norteks' software quality control. However, the data were complemented by quality indexes based on detailed analysis of observations.

### 3.1 Data integrity and correction

Overall, the observations presented here cover a period equivalent to 4.9 consecutive years, between 14-November-2016 and 30-August-2022. As to the writing of this paper, the DeepLev operation is still ongoing, the ninth deployment of the wave monitoring ADCP which began on January 2023 and recovered in the beginning of 2024 (It will be analysed and added to the dataset when ready). Table 2 describes the data obtained from each deployment along with assigned quality indexes. Predictably, the majority of observations are of good quality and provide the full set of wave characteristics including directional information. Soffer et al. (2020) previously compared wave parameters from the DeepLev's first deployment with a simultaneous measurement of a bottom-mounted ADCP which was located 48.5 km away at a depth of 26 m. Both presented a stormy event with reasonable differences given the distance between the locations providing initial validation to the reliability of "Signature-500" measurements from the subsurface buoy. However, in some of the later deployments we have faced several challenges during data processing and analysis. Some of them were resolved and others are yet to be explained.

The initial challenge we encountered was a considerable variability in the percentages of "Ambiguous" data indicating the inability of the system to determine a local maximum of the wave energy spectra. Naturally, The situation occurs more

frequently while the nominal depth of the buoy carrying the ADCP is higher. When installing a moored station with 1470 m long cable, it was difficult to ensure the precise depth of the sub-surface buoy. In practice, the nominal depths varied by 12 m (27-39 m), therefore some deployments retrieved higher percentages of directional data than others. The analysis showed that for the specific wave characteristics measured, securing the instrument at 30 m bellow the sea surface would add another $10\%$ of valid data to the gathered wave directional information.

Only a small portion of the measurements were found to be unreasonable or completely missing. Occasionally, if there is a problem with returning bursts or if the device has trouble detecting the surface it will lead to missing points after processing. Unfortunately, two of the deployments (the fourth and the eighth) had issues resulting in abnormal data loss. During the fourth deployment, it seems like something obstructed the device as evidenced by notable deviations between the measured distance and pressure. A relatively short timeseries of the eighth deployment stem from an unexpected malfunction of the memory card.

Another problem was addressed after the initial processing. In both, the second and third deployments, the instrument returned without the ordinary temperature readings. Normally, this information is used to evaluate the water's sound velocity (SV) which is necessary to translate the return time of a burst to distance. As a consequence of the fault the initial processing for these deployments was carried out with the Nortek's software default SV value of 1300 m $\sec^{-1}$. In practice, the appropriate values for the water properties in that area are around 1550 m $\sec^{-1}$. This means that the calculations were performed with SV values lower by about 20% which led to similar deviations in computed length scales. To correct these values, the missing temperatures were replaced with records by a secondary temperature sensor attached to the pressure sensor. Using these data, the SV was recalculated with the Gibbs-SeaWater (GSW) Oceanographic Toolbox (McDougall and Barker, 2011). Then the calculated parameters were adjusted according to the ratio between the new SV and the original ones. A good indication that the correction succeeded was to compare the adjusted distances from the AST measurement and to the pressures observations. After the adjustment the two series differed from each other in the same manner as in the remaining deployments. In this regard, one should consider that the SV used for calculations is constant even if the water column is strongly stratified. As it happened during the local summers when according to the temperature measurements the thermocline was located above the ADCP. Then, the assumption that the measured values fitted the entire water column turns out to be inaccurate. In such a case the calculations are based on a temperature measured bellow the thermocline while the water column between the device and the surface are likely $8°C - 10°C$ warmer. As a result, the uncertainties in SV and wave height estimates could reach $2 - 3\%$.

Lastly, observing waves from a submerged subsurface buoy adds complexity since the measurements are caring out from a moving platform. The processing software uses records from the tilt sensors for corrections. But, to get a good reading from the AST sensor the tilt must be lower than $10°$. With specified DeepLev station mooring settings, there were no instances of angles exceeding this value with the maximal registered tilt reaching $8°$. Additional variability manifests in the horizontal and vertical location of the subsurface buoy which mostly caused by the forces the flow applies on the entire mooring system. At the most extreme case the buoy descends by 30 m in 6 h meaning it does experience occasionally substantial changes in depth even within the 17 min windows we use for analysis. These movements can have a slight impact on the quality of the measurement as it depicts an average across varying conditions but for the most part is negligible since a linear detrend is performed prior to wave parameter extraction. Another type of buoy motion is its response to the surface waves. According to

the instrument's accelerometer record of the fourth deployment, the buoy experiences horizontal movements which resemble the frequency distribution of surface waves with a peak around 0.1 Hz. Whereas, the vertical accelerations' distribution presents as symmetric centered at 0.125 Hz (8 sec wave period) hinting a resonant response, likely a buoyancy-related natural harmonic. Surface wave components around this frequency regularly induced sway in order of magnitude just a few to tens of centimeters which could add bias or random error to the directional estimates. Encouragingly, this motion is not substantial as appears in Pedersen et al. (2007), probably because the installations were deeper and the natural frequencies were higher.

## 3.2  Data review

The developed dataset represents an open source of surface wave characteristics derived from ADCP measurements (https://doi.org/10.17882/96904). The number of files corresponds to the number of deployments which simplify the selection of the timeseries of interest. The used NetCDF4 format guarantees easy access and eliminates occasional reading errors. Each file contains the time varying spectra $S^{ast}(f)$, $S^{vel}(f,\theta)$ and $S^{suv}(f,\theta)$. In addition, it includes unified arrays of the aforementioned statistical wave parameters with preference to values derived from $S^{suv}(f,\theta)$. The frequency range for wave spectra is 0.02-0.45 Hz with the step of 0.005 Hz. A few isolated events led the ADCP to experience deepening of over 10 m. The maximal recorded depth was 54 m on 20-March-2022, thus lowering the frequency ambiguity limit to 0.195 Hz. A full description of the files with detailed specification of each wave parameter is available in the appendix.

Figure 2 shows a timeseries of $H_{m0}$ and $T_p$ reconstructed by two methods (VEL and AST) for a short period out of the 3rd deployment. This time frame includes the highest observed waves event of the entire campaign when $H_{m0}$ reached 8 m. Apparently, when the surface waves are high and long there is a good agreement between the two methods. The preference of using the AST approach is eminent in young waves conditions (fig 2b). When it comes to directional spectra, the ability of ADCP is limited in very rough sea state, so the example for retrieved spectra is taken after the peak of the event (Shown in Fig. 4). Both methods demonstrate a consistency in directional distributions. The incorporation of the AST in the SUV method adjusts the intensities and energy distribution between frequency bins.

Figure 3 displays the distributions of $H_{m0}$, $H_{max}$, $T_p$ and $T_{m02}$ among all the data collected. Though there are gaps between deployments, all months were sampled fairly evenly so the results are not expected to be strongly biased. The most probable wave statistics at the DeepLev location have $H_{m0}$ between 0.5 m and 1 m and a $T_p$ of 5-6 sec. Moreover, at least half the time the $H_{m0}$ is over 0.8 m and finding it measuring up to 2.5 m with $H_{max}$ of 4 m is common. To give additional overview of the measured wave distributions, observations between 14-November-2016 and 30-June-2021 are compared with model results from the Copernicus Marine Environment Monitoring Service (CMEMS) which implements the WAM model (Günther et al., 1992; Komen et al., 1996) to simulate waves in the Mediterranean Sea[1]. Figure 5 show the density scatter plots for $H_{m0}$, $T_p$ and $T_{m02}$ with the corresponding Pearson correlation coefficient, bias, root mean square error (rmse) and scatter index (si). The statistical estimators for $H_{m0}$ are comparable to the results of Coppini et al. (2023) that validated CMEMS-WAM using several buoys around the Mediterranean coasts. The correlation coefficient is high, while the other values fall within the range of buoys. When comparing to their average values of all buoys the bias and rmse are more significant. This can be attributed to

---

[1]https://doi.org/10.25423/cmcc/medsea_multiyear_wav_006_012

the bias of forcing winds in the Eastern Levant as seen in their comparison to satellite altimetry or the measuring methodology of using an ADCP mounted on a subsurface buoy. Comparison of $T_{m02}$ in Fig. 5c shows a general good trend but with a negative bias between modeled and observed values. This bias is not reflected in the $T_p$ scatter which is concentrated around the best fit diagonal meaning the contribution to the bias is mostly caused by instrument's limit of measuring short waves. Logically, other calculated parameters, like $H_{m0}$, could also be affected by the lack of short waves but as these waves are typically less energetic the influence is not accentuated. Nonetheless, when working with spectral data it is recommended to integrate all parameters of interest only within the instrument's resolved frequency range for optimal comparisons.

## 4    Summary and Conclusion

Wind wave characteristics have been assembled together after multistage data processing, correction, and analysis for an extended period between 2016-2022. The developed dataset derived from an Acoustic Doppler Current Profiler is a part of the comprehensive DeepLev monitoring project in the Levantine basin off the Israeli shore. The analyzed data constitute a timeseries of full two-dimensional wave fields, calculated by two methods utilizing wave orbital velocities and surface tracking, along with conventional statistical parameters: wave heights, periods, and directions of propagation. Preliminary statistical analysis were performed to showcase the distributions, medians and maximal values of principle wave parameters. Finally, a comparison of observed significant wave heights and mean wave periods to parallel model values shows a gap in estimated periods and an underestimation in modeling of high waves.

Such a valuable add-on to the exploring of the Levantine Sea is of importance considering the deficiency of observations compared to other sub-basins of the Mediterranean Sea. The collected data can be effectively used for monitoring wave climate changes on seasonal and long term scales as well as for evaluation of extreme wave characteristics or wave energy in the Eastern Mediterranean. Beyond the importance of the dataset to this specific region, it is an uncommon extensive timeseries of deep water wave spectral measurements which can generally contribute to marine studies. Besides scientific findings, this experiment also have brought valuable insights on long exploitation of the ADCP Nortek "Signature-500" in deep waters.

To finalize the paper, we would like to stress the value and importance of a unique five year dataset of wave characteristics in the deep waters of the Eastern Mediterranean basin for sea state monitoring.

*Data availability.*

Described data are freely available through SEANOE (SEA scieNtific Open data Edition) open scientific data repository: https://doi.org/10.17882/96904 (Haim et al., 2022).

*Author contributions.* NH managed processing and organization of data, NH, VG, and YT wrote the manuscript, analysed, evaluated and visualized the processed data. YT, RS and BM maintained the "Signature-500" instrument and its operation. The authors TK, RA, EB, AL, HG, IB-F, YW and BH contributed via their labor in the long-term ongoing DeepLev project, its maintenance, operation and management. Funds were raised by IB-F, IW, BH and YT

*Competing interests.* none

*Acknowledgements.* This research was supported by the ISRAEL SCIENCE FOUNDATION [grant No. 25/2014 (IW); and 1940/14 and 1601/20 (YT)] in frame of the used equipment and the student grants. We are grateful to the Council for Higher Education in Israel and the Mediterranean Sea Research Centre of Israel (MERCI), the Wolfson Foundation, the North American Friends of IOLR and Bar-Ilan University (BIU) for funding the construction and maintenance of DeepLev. This project was supported by the Israeli Ministries of Energy
and Environmental Protection under the framework of the National Monitoring Program for Israeli Mediterranean Waters. We would like to show our appreciations to the IOLR electronic, sea operations and marine physical departments for invaluable help with facilities and technical operations in establishing and running the DeepLev station, the captain and crew of R/V Bat-Galim, and the engineers of the machine shop at BIU for their help in the buildup of mooring.

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

| # | Deployment start | Deployment end | Duration [days] | Sampling Frequency [Hz] | Interval [min] |
|---|---|---|---|---|---|
| 1 | 14-Nov-2016 | 12-May-2017 | 179 | 2 | 120 |
| 2 | 1-Jun-2017 | 25-November-2017 | 177 | 2 | 120 |
| 3 | 4-Dec-2017 | 28-April-2018 | 145 | 2 | 17* |
| 4 | 31-Jul-2018 | 28-March-2019 | 240 | 4 | 17* |
| 5 | 13-May-2019 | 18-December-2019 | 219 | 2 | 120 |
| 6 | 18-Feb-2020 | 16-September-2020 | 211 | 2 | 60 |
| 7 | 27-Oct-2020 | 3-November-2021 | 372 | 2 | 60 |
| 8 | 27-Dec-2021 | 30-August-2022 | 246 | 2 | 60 |

**Table 1.** Duration for each of the ADCP deployments with measuring configuration: Sampling frequency of sensors and interval of measurements in "Burst" mode. *"Continuous" mode measurements which were then processed in 17 min windows.

| # | Nominal depth [m] | Time points | Valid [%] | Ambiguous [%] | Unreasonable [%] | Missing [%] |
|---|---|---|---|---|---|---|
| 1 | 39 | 2153 | 82.86 | 15.00 | 1.90 | 0.23 |
| 2 | 31 | 2128 | 97.70 | 1.13 | 1.08 | 0.09 |
| 3 | 32 | 12235 | 88.39 | 7.61 | 3.95 | 0.06 |
| 4 | 28 | 20240 | 84.90 | 1.93 | 0.89 | 12.29 |
| 5 | 29 | 2621 | 92.79 | 4.85 | 2.37 | 0.00 |
| 6 | 39 | 5066 | 82.23 | 12.34 | 2.33 | 3.10 |
| 7 | 37 | 8921 | 84.92 | 12.70 | 0.91 | 1.47 |
| 8 | 27 | 5905 | 32.35 | 2.64 | 1.61 | 63.40 |

**Table 2.** Summary of quality indexes for each deployment as described in Sec 3.2.

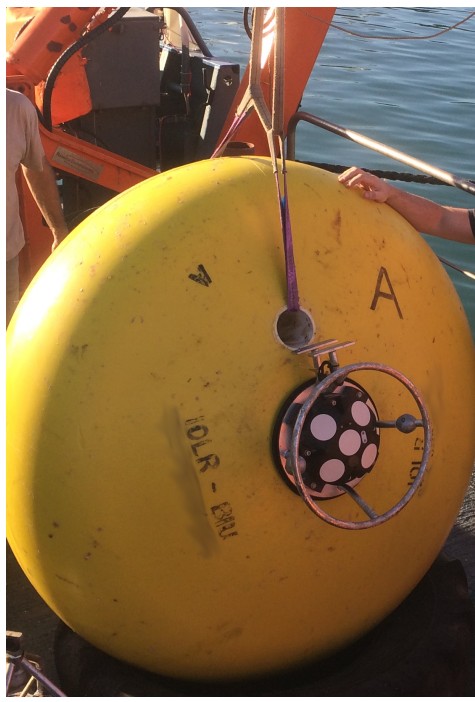

**Figure 1.** The wave measuring instrument, Nortek's "Signature-500", mounted on the top buoy of the "DeepLev" mooring system.

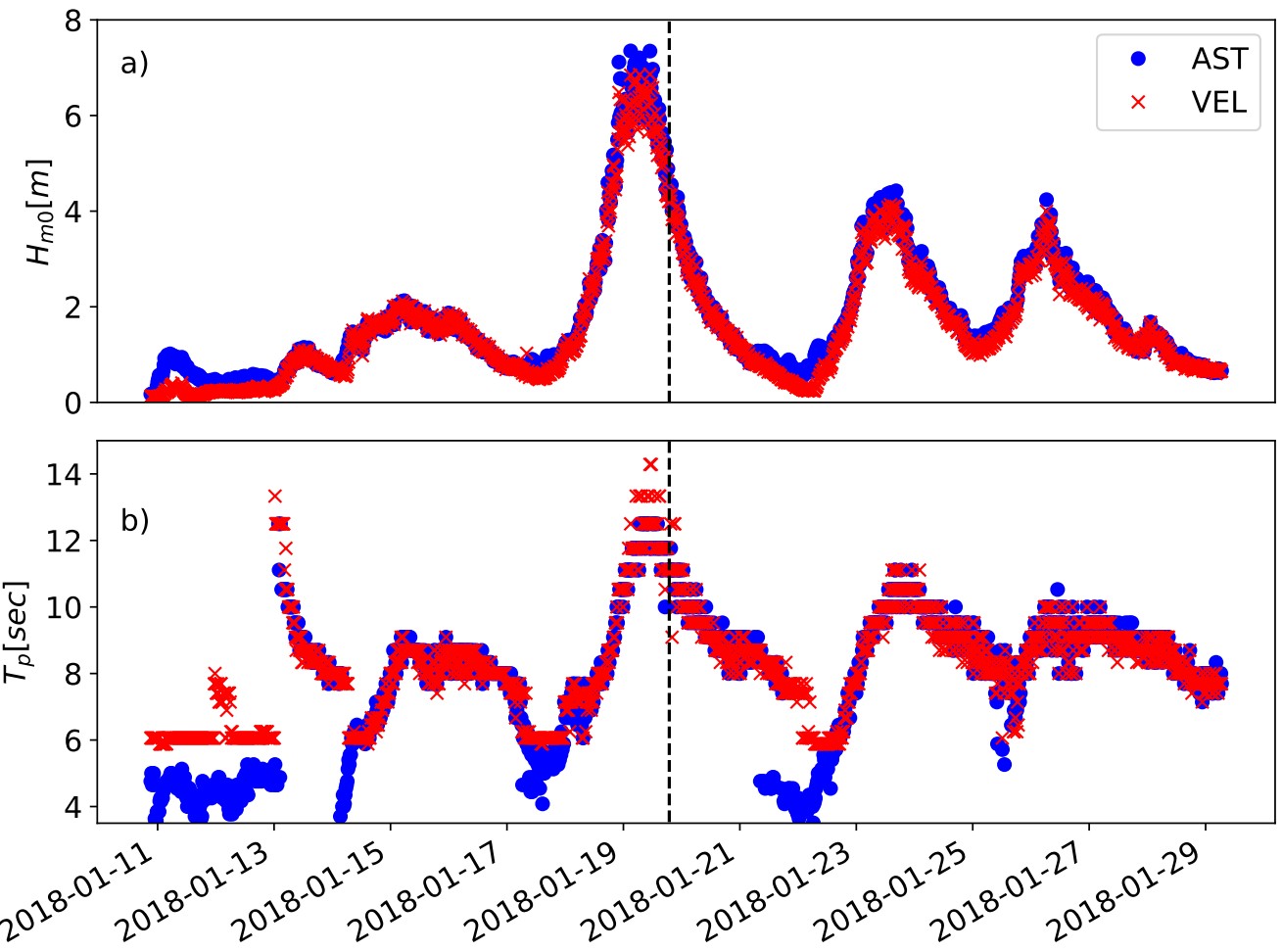

**Figure 2.** A short timeseries out of the 3rd deployment derived from velocity orbitals (red) or combined with AST (blue)). Shown parameters are a) Significant wave heights and b) peak period. Vertical dashed line marks the date of measurements presented in Fig. 4

.

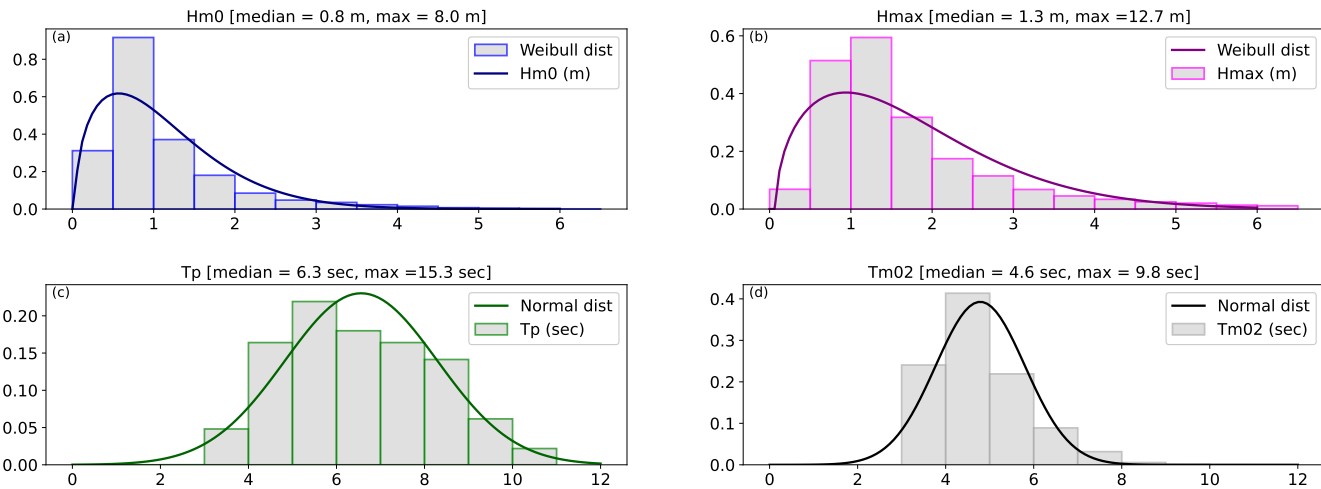

**Figure 3.** Histograms of combined data from the 8 deployments. a) significant wave heights $H_{m0}$ b) maximal wave heights $T_{max}$ c) peak wave periods $T_p$ d) mean wave periods $T_{m02}$. Accompanied by approximated probability fits for comparison.

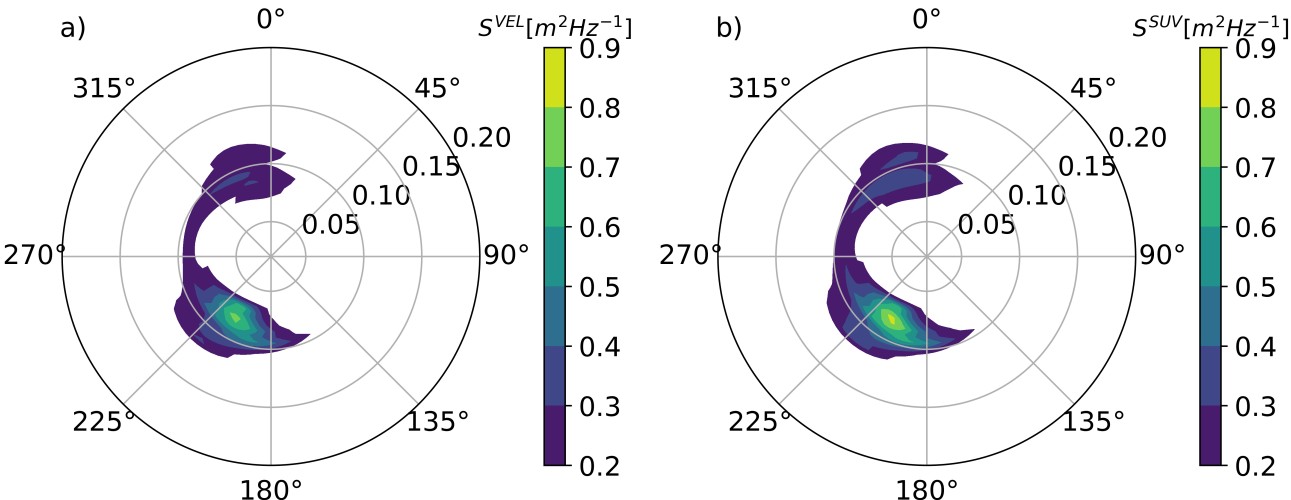

**Figure 4.** Directional energy density spectra observed on 19-January-2018 18:54 processed from a) velocity orbitals, $S^{vel}(\theta, f)$, or b) combined with AST, $S^{suv}(\theta, f)$.

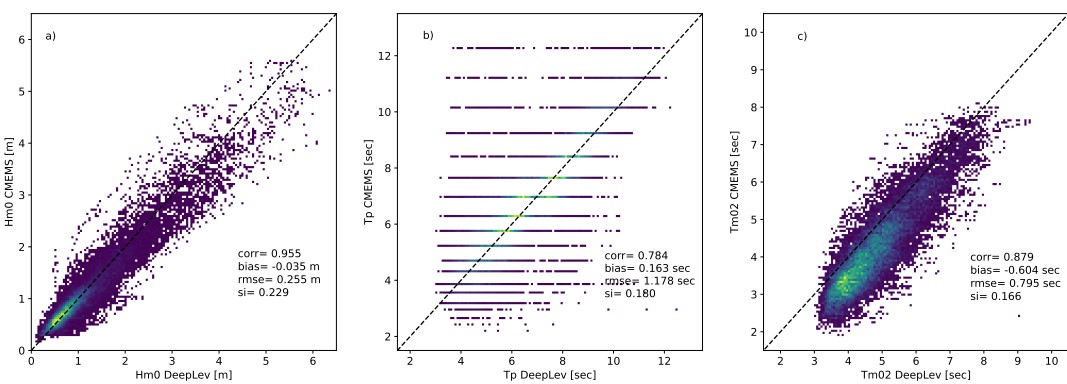

**Figure 5.** Density scatter plots of the observed a) $H_{m0}$, b) $T_p$ and c) $T_{m02}$ values to modeled values from CMEMS WAM between 14-November-2016 and 30-June-2021.

## Appendix A

| Parameter Name | Notation | Dimensions | Description |
|---|---|---|---|
| Direction | | frequency, time | Dominant direction of each frequency component |
| WaveSpectra AST | $S^{ast}(f,t)$ | frequency, time | Spectral analysis of AST |
| WaveSpectra Pressure | $S^{p}(f,t)$ | frequency, time | Spectral analysis of pressure |
| WaveSpectra Vel | $S^{vel}(f,t)$ | frequency, time | Analysis of surface velocity magnitude |
| EnergySpectra | | frequency, time | Compilation of $S^{ast}, S^p$ and $S^{vel}(f,t)$ |
| FrequencyAmbiguityLimit | | time | Cut off frequency for directional analysis |
| VelocitySpectra_Energy | $S^{vel}(\theta, f, t)$ | time, direction, frequency | VEL method, uses velocities |
| ASTSpectra Energy | $S^{suv}(\theta, f, t)$ | frequency, time | SUV method, uses AST and velocities |
| PressureSpectra_Energy | $S^{puv}(\theta, f, t)$ | time, direction, frequency | PUV method, uses pressure and velocities |
| FullWaveDirectionalSpectra_Energy | | time, direction, frequency | Compilation of $S^{suv}, S^{puv}$ and $S^{vel}(\theta, f, t)$ |

**Table A1.** 1D and 2D spectral energy densities included in the Netcdf files. More details on the analysis methods can be found in Section 2.

| Parameter Name | Notation | Description | Units |
|---|---|---|---|
| Temperature | | averaged temperature | C |
| Tilt Pitch | | | degree |
| Tilt Roll | | | degree |
| Heading | | | degree |
| Pressure | P | averaged water column pressure | dbar |
| Distance | | distance from surface measured by vertical acoustic beam (AST) | m |
| Current Direction | | | degree |
| Current Speed | | | m/sec |
| Direction DirTp | $\theta_p$ | Direction at peak wave period | degree |
| Direction MeanDir | $\theta_m$ | Mean Direction | degree |
| Direction SprTp | | Spreading at peak wave period | degree |
| Height H10 | $H_{10}$ | mean height of the 10% largest waves (observed by AST) | m |
| Height H3 | $H_3$ | mean height of the 33% largest waves (observed by AST) | m |
| Height Hm0 | $H_{m0}$ | spectral significant wave height | m |
| Height Hmean | $H_{mean}$ | mean height of all surface waves (observed by AST) | m |
| Height Hmax | $H_{max}$ | Highest single wave height (observed by AST) | m |
| Period T10 | $T_{10}$ | mean period of the 10% largest waves (observed by AST) | sec |
| Period T3 | $T_3$ | mean period of the 33% largest waves (observed by AST) | sec |
| Period Tenergy | $T_{energy}$ | $m_{-1}/m_0$ | sec |
| Period Tm02 | $T_{m02}$ | $\sqrt{m_0/m_2}$: spectral mean wave period | sec |
| Period Tmax | $T_{max}$ | wave period of single largest wave (observed by AST) | sec |
| Period Tp | $T_p$ | $1/f_p$ wave period of peak wave frequency | sec |
| Period Tz | $T_z$ | Mean zero-crossing wave period | sec |
| SpectrumType | | origins of values in 1d spectral variables 0: Pressure 1:Velocity 3:AST | |
| ZeroCrossings | | number of zero crossings | |
| QI | | quality index as desribed in section 3. 1: valid, 2: ambigious, 3: unreasonable, 4:fault | |

**Table A2.** Timeseries of wave parameters and sensors records included in the Netcdf files. Parameter Names as appear in the files and notations as appear in the text.