# Peer review of "Multiyear surface waves dataset from the subsurface "DeepLev" Eastern Levantine moored station"

_Earth System Science Data, 2023_

## Author Response (AR1)

**Reviewer 1**

**This is a clean paper describing a dataset made publicy available. The layout and presentation are essential, clear, sufficient (but see my notes below), so, for what it is, the paper, basically an interesting report, is acceptable. Certainly, using an ICTP on top of a 1400 m depth mooring is a non-trivial challenge. The related problems are mentioned, although it would have been interesting to go a bit more into the related details (see below). Specific comments moving along the paper.**

1) **- Abstract and also text – In my opinion the measurements are interesting in themselves without the need to invoke wind wave generation mechanisms, climate change et al.**

Thanks for this comment. The last sentence of the abstract as edited accordingly:

"The developed long-term timeseries of wave parameters contribute to monitoring and analysis of the region's wave climate, and for evaluating the quality of wind-wave forecasting models"

Changes applied to the abstract (line 10) and Section 4 (lines 207-208)

2) **- l(ine)28 – As stated, the sentence appears to claim all this has been done for the first time. I believe this is not the case. The last five words "for the period 2016-2022" make the overall claim ambiguous. I would rephrase it.**

We agree, the phrasing implied this was the first installation of subsurface buoy for waves measurements. It was not our intention to suggest that this work was the first to deploy a deep sea mooring for wave monitoring. The sentence was referring to the DeepLev station as its first of a kind multi-disciplinary platform for marine observations specifically in the Levant to emphasize the importance of this project. We rephrased the sentence accordingly.

Changes applied to Section 1 (line 28-30)

3) **- l40 – my feeling is that the buoy wave induced movements are not "minor data artefacts".**

The sentence was a reference to Pedersen et al., 2007 which discussed artifacts in their subsurface ADCP experiments. To avoid any misinterpretation of their results we removed the word "minor" following your comment. Later we also discuss the influence of buoy motion on our measurements.

Changes applied to Section 2 (line 50)

4) **- l44 - "ADCP, allowing". Otherwise it looks like it is the ADCP that allows.**

Thank you. Changed as suggested.

Changes applied to Section 2 (line 54)

**5)  - l43-58 – I suggest a table summarizing acronym, which sensors are used in each approach, limits.**

Table A1 was already summarizing the different types of spectra available. It was edited to include which sensor is used in each approach.

Changes applied to Appendix table 1

**6)  - l57-58 – it depends on which waves (storms or else) you are interested in.**

That is true, we rewrote the sentence to clarify that in our installation most of the wind-sea spectrum is not detectable by pressure but the lower frequencies such as swell could appear in the data.

Changes applied to Section 2 (line 67-70)

**7)  - l79 – Usually the expression "one-dimensional frequency spectra" is used. Not essential.**

Thanks, we changed to just "frequency-specra".

Changes applied to Section 2 (line 62)

**8)  - l80 – I do not remember the exact figure, but usually Hm0 and H3 have a well defined difference.**

Thanks, we removed the sentence comparing the two to avoid confusion.

Changes applied to Section 2 (line 92)

**9)  - l90 – "provides …. allows"**

The sentence was edited as suggested.

Changes applied to Section 2 (line 102)

**10) - l99 – Given that the 8th period ended 15 months ago, I assume this sentence needs to be rephrased**

The 9th was deployed late and still currently at sea, planned to be retrieved in the coming month. Therefore the phrasing "As to the writing of …" is fitting and is kept.

Changes applied to Section 3 (line 102)

**11) - l102-103 – Not clear. Some more details would be useful**

Thank you for the comment. The purpose was to give some assurance that the data included are reliable. Soffer et al 2020 previously compared wave parameters from the DeepLev's first deployment with a simultaneous measurement of a bottom mounted ADCP which was located 48.5 km away at a depth of 26 meters. Both presented a stormy event with reasonable

differences given the distance between the locations providing initial validation to the reliability of Signature-500 measurements from the subsurface buoy.

Changes applied to Section 3 (lines 114-118)

**12) - l104 "others are yet"**

Thanks, change as suggested

Changes applied to Section 3 (line 114)

**13) - l107 – not clear to me the words "resulted of"**

Thanks, was rephrased for clarity.

Changes applied to Section 3 (line 123)

**14) - l108-109 – depth varying by 12 m. Pretty interesting. This points to a more general problem. Obviously in stormy conditions the 30 m depth buoy must move quite a bit, especially horizontally. This must have consequences on the measurements. In my view this is not sufficiently discussed.**

That is true, it should be further discussed in the paper. Regarding the location and motion of the buoy there are two separate matters. First, there is some variability in positioning between each deployment that manifests in uncertainty of spatial location and in the measured nominal depth that varied by 12 m. The horizontal and vertical location obviously changes with the currents where at the most extreme case the buoy descends by 30 meters in 6 hours meaning it does experience occasionally significant changes in depth even within the 17 minutes windows we use for analysis. The second type of motion is with the waves, certainly we see their influence in the accelerometers. In terms of positioning this effect only causes changes in magnitude of centimeters and the Nortek software consider buoy motion in the analysis. Following your comments, we are currently looking into the natural frequency of the buoy which can interrupt the observation. But it seems that its oscillations are centered around 8 seconds which is short enough so that the surface waves will not have an immense effect at 30 meters.

Changes applied to Section 3 (line 155-165)

**15) - l120 – Either I do not understand or something is wrong. You take 1300 m/s as default value of water's sound velocity. The sensitivity to temperature is about 4.5 m/s/deg C. So where does the 20% come from? Also 10 deg difference suggest 45 m/s difference, i.e. the 2-3% then mentioned at line 130.**

Apologies, the source of the 20% error was not clear enough. The value of 1300 is the software's default while true values should be 1530-1580 m/sec. That is the origin of the 20% error of the initial processing. This difference was corrected as mentioned in the manuscript by using temperature records of the pressure sensor. Indeed, this 45 m/sec difference for a value in the 1500 m/sec range introduces an additional 2-3% which we cannot address.

Changes applied to Section 3 (line 136-138)

**16) - l125 – SV?**

No, it is indeed AST. The sentence was rephrased for clarity.

Changes applied to Section 3 (line 143-145)

**17) - l140-145 – frequency limits. Are these due to attenuation with depth  (but at 30 m depth the 0.445 Hz waves are virtually 0) or else? My ignorance perhaps, but I could be not alone.**

The AST is not limited by wave's depth attenuation as it measures the reflection from the surface itself. It is limited by the widening of the acoustic beam with distance but theoretically 0.445 Hz oscillations are still detectable by the AST. Practically, only a handful of times the instrument got readings at 0.45 Hz, that is why the limit was adopted.

Changes applied to Section 3 (line 171)

**18)  - l158-159 – This leads me back to the note on l28.**

Please see our answer at the comment for l28.

Changes applied- see comment #2

**19) - l166 – I do not see how these data can be informative on wave generation, growing and decay. As already said, measured data are valuable in themselves.**

Thank you, we agree the data is valuable by itself. Invoking wave-generation was a mistake on our behalf. We still wish to suggest that the data can contribute to study of local wave climate and in validation of wave models as it is a long period observation at deep sea which is not very common.

Changes applied- see comment #1

Reviewer 2

The ms is quite interesting since it discusses new collected in-situ wave data that was obtained over several years in the SE Levantine basin offshore Israel using an ADCP, which was used for the first time at the area of interest's subsurface water layer.

1) To give confidence for their future use and applications, the dataset must be assessed with other independent data, even with a thorough explanation of the data processing methods and the many issues encountered during data collecting.

   Therefore, I propose the authors to undertake a basic statistical evaluation of their wave data parameters using the available offshore Israel wave data covering the same in-situ period, i.e., those of the CMEMS Med MFC. Moreover, will be of useful to the ms to make the same evaluation using the in-situ data from the nearby Hadera wave station.

2) Additionally, the authors ought to consult earlier studies addressing wave characteristics offshore the eastern Levantine basin for the sake of inter-comparison; one such study that comes to mind is the one by Zodiatis G., G. Galanis, G. Kallos, A. Nikolaidis, Chr. Kalogeri, Aris. Liakatas and S. Stylianou (2015). The impact of sea surface currents in wave power potential modeling. Ocean Dynamics, 65:1547, DOI: 10.1007/s10236-015-0880-4.

Thank you for your interest in our manuscript and supportive review. We decided to focus this paper on the published dataset. Nonetheless, we understand the contribution of a comparison to a wave model and accordingly added a brief comparison to CMEMS's model as suggested. Our comparison shows low bias (-0.025m) and rmse (0.25m). A discussion on this comparison was added to the end of Section 3. Comparing to distant in-situ measurement such as the ones in Hadera should result in differences that require extensive analysis (see for example Soffer et al. 2020) which are out of the scope of this paper.

Changes applied to Section 3 (line 187-195)

We added relevant references to the earlier studies of Zodiatis et al. (2014), Zodiatis et al.(2015) Coppini et al (2023) in the introduction addressing wave characteristics and wave power potential in the Eastern Levantine Sea.

More references were added to Section 1 (line 11-15 and lines 23-25)

More Changes were applied to the conclusion in Section 4 in accordance with the overall changes made to the manuscript

---

## Referee Report (RR1)

Review of

**"Multiyear surface waves dataset from the subsurface 'DeepLev' Eastern Levantine moored station"**

by Haim et al.

This is a valuable paper and dataset no doubt worthwhile of publication. I have some notes and suggestions along the paper that I list below.

**l(ine)13** – It is not because I am one of the co-authors, but I believe that the timeseries reported by Pomaro et al (since 1979) should be mentioned.  Reference:

2018  A.Pomaro, L.Cavaleri, A.Papa, P.Lionello, "39 years of directional wave recorded data at the Acqua Alta oceanographic tower" *PANGAEA*, https://doi.org/10.1594/PANGAEA.885361,

**l27** et al – I do not have a solution, however my feeling (possibly biased by the one in front of Venice) is that usually peole mean something different with the word 'platform'

**l91** – My opinion is that it would be correct to mention the original paper by Longuet-Higgins, Cartwright and Smith

**l101** – "... all measurements that passed ..."

l**107** – Is this deployment still going? If not, it should be mentioned

**l109-110** – 'Soffer et al, 2020' is mentioned twice. Previously …

**l150** – My feeling is that possibly the situation is slightly more complicated. Ok, you detrend, but this means that the depth changes during the record, and this should affect the attenuation, the measurements, hence the estimate of the wave parameters. Am I correct?

**l152** – natural period of the buoy. For which motion?

**l177 –** Ok for Gunther et al, 1992, but I believe the standard reference for the WAM model is Komen et al 1994 (again, I am not pressing because I am one of the authors)

**l180** – I am well aware of the wind bias in the Mediterranean Sea, but in my opinion some more details are required. Otherwise it aears as an excuse.

**l197** - "... this specific region, it is ..."

**Figure 5** – right panel. In my opinion it would be interesting and instructive to extend the lower limit of the two axes to lower T values. There is the obvious problem of the attenuation of waves with depth, especially when waves are shorter (lower periods). In any case the apparently lower general periods of the model, also for longer periods, is not fully consistent with what shown in the left panel.

Nice work and dataset obtained in difficult conditions with an innovative approach.

Luigi Cavaleri

---

## Referee Report (RR2)

**General Comments:**

The paper describes a new data set of 2D wave fields derived from the analysis of in-situ measurements collected from an ADCP mounted on a subsurface deep mooring deployed for first time at the Eastern Mediterranean, 50km of the Israeli coast, west of Haifa, for the period 2016-2022. The methods used for the processing, correction, and analysis of the data are analytically described. The in-situ observations were evaluated by comparison with the outputs of the Copernicus Marine Environment Monitoring Service (CMEMS) wave (WAM) model.

Such a time series of wave data is very important for the area and the Eastern Mediterranean in general as it is a region of increased scientific and economic interest.

The paper can be accepted for publishing, below are some issues to be addressed before the publication.

**Comment on the data availability:**

The paper states that the data are freely available through SEANOE repository (https://doi.org/10.17882/96904). However, currently the access is not open and an embargo period is set until 1-11-2025, which is in contrast with the paper. I would suggest authors remove the "freely" from the data availability section, add that an embargo period exists and explain why.

**Specific comments:**

(lines numbering corresponds to the Author's tracked changes file)

- Line 114: sentence is not clear to me, "Soffer et al. (2020)" is repeated twice. A suggestion could be: In a recent study, Soffer et al. (2020) compared wave parameters from the DeepLev's first deployment with simultaneous measurements from a bottom-mounted ADCP located 48.5 km away at a depth of 26 meters.
- line 137: Although I am not a native English speaker, I would suggest this sentence as: "In practice, the appropriate values for the water properties ..."
- Table 2 caption: I would change the caption from "Summary of quality of data ..." to "Summary of quality indexes of data ..." or "Statistics on quality of data ..."

**TEXT editing:**

- There is an inconsistency in figure references that should be fixed. Base on the guidelines (https://www.earth-system-science-data.net/submission.html#figurestables) it should be (Fig.2b) and not (fig 2b), or (Fig. 4) and not (Figure 4), etc.
- the "meter" unit is not used in a consistent way, e.g. line 117: 26 meters, line 125: 12 m (27-39 m). Should be fixed.

---

## Author Response (AR2)

Reviewer 1

Review of
"Multiyear surface waves dataset from the subsurface 'DeepLev' Eastern Levantine moored station"
by Haim et al.
This is a valuable paper and dataset no doubt worthwhile of publication. I have some notes and suggestions along the paper that I list below.
l(ine)13 – It is not because I am one of the co-authors, but I believe that the timeseries reported by Pomaro et al (since 1979) should be mentioned. Reference:
2018 A.Pomaro, L.Cavaleri, A.Papa, P.Lionello, "39 years of directional wave recorded data at the Acqua Alta oceanographic tower" PANGAEA,
https://doi.org/10.1594/PANGAEA.885361,

Thank you for highlighting this work. We agree it is an important example where we mention waves monitoring in the Mediterranean Sea.

Change applied to line 13

l27 et al – I do not have a solution, however my feeling (possibly biased by the one in front of Venice) is that usually peole mean something different with the word 'platform'

To improve readability we changed the first two instances when we describe the DeepLev station as a platform. We use the term later only as a part of the phrase "moving-platrorm"

Change applied to lines 26 and 42

l91 – My opinion is that it would be correct to mention the original paper by Longuet-Higgins, Cartwright and Smith

A cititation was added

Change applied to line 81

l101 – "... all measurements that passed ..."

Thank you, the text was edited.

Change applied to line 100

l107 – Is this deployment still going? If not, it should be mentioned

The 9th deployment was recovered but not yet processed. It will be added to the dataset in the future (now mentioned in the paper). The Project is still going.

Change applied to line 106

l109-110 – 'Soffer et al, 2020' is mentioned twice. Previously …

Thank you, the sentence was fixed.

Change applied in lines 109-110

l150 – My feeling is that possibly the situation is slightly more complicated. Ok, you detrend, but this means that the depth changes during the record, and this should affect the attenuation, the measurements, hence the estimate of the wave parameters. Am I correct?

The computed spectra and accompanying parameter are always averaged representations within the chosen window. The question is whether the conditions change too drastically so that the averages make no sense. The maximal experienced change within 17 minutes is 1-2 m which does not impact too much the cut-off frequency. We added a comment in the text to address it.

Change applied to lines 148-149

l152 – natural period of the buoy. For which motion?
To answer this question, we extended the description of accelerometer records analysis. It gives different distributions for vertical and horizontal movements. In both x and y directions the acceleration spectra reminds the distribution of surface waves. While the vertical component is entirely different, it is symmetric and centred around a frequency of 0.125Hz. Such a complicated system could have many natural frequencies but the dominant feature in the vertical accelerations is likely buoyancy related.

Change applied to lines 150-153

l177 – Ok for Gunther et al, 1992, but I believe the standard reference for the WAM model is Komen et al 1994 (again, I am not pressing because I am one of the authors)

Thank you, added the suggested reference.

Change applied to line 179

l180 – I am well aware of the wind bias in the Mediterranean Sea, but in my opinion some more details are required. Otherwise it aears as an excuse.

You are absolutely right, it is not the only cause and the paragraph was re-written to address this comment and the last one regarding the scatter plots.

Change applied to lines 182-190

l197 - "... this specific region, it is ..."

Thank you, the comma was added.

Change applied to line 203

Figure 5 – right panel. In my opinion it would be interesting and instructive to extend the lower limit of the two axes to lower T values. There is the obvious problem of the attenuation

of waves with depth, especially when waves are shorter (lower periods). In any case the apparently lower general periods of the model, also for longer periods, is not fully consistent with what shown in the left panel.

The ranges of figure 5 were adjusted as suggested. In addition, we added a third scatter plot of Tp to solidify that the biases are due to this attenuation and not a general bias in the observed wave period interpretation.

Change applied to lines 182-190

Nice work and dataset obtained in difficult conditions with an innovative approach.
Luigi Cavaleri

Thank you for investing time and effort. Your questions and comments were insightful.

Reviewer 2

General Comments:

The paper describes a new data set of 2D wave fields derived from the analysis of in-situ

measurements collected from an ADCP mounted on a subsurface deep mooring deployed for

first time at the Eastern Mediterranean, 50km of the Israeli coast, west of Haifa, for the period

2016-2022. The methods used for the processing, correction, and analysis of the data are analytically described. The in-situ observations were evaluated by comparison with the outputs of the Copernicus Marine Environment Monitoring Service (CMEMS) wave (WAM) model.

Such a time series of wave data is very important for the area and the Eastern Mediterranean

in general as it is a region of increased scientific and economic interest.

The paper can be accepted for publishing, below are some issues to be addressed before the publication.

Comment on the data availability:

The paper states that the data are freely available through SEANOE repository

(https://doi.org/10.17882/96904). However, currently the access is not open and an embargo period is set until 1-11-2025, which is in contrast with the paper. I would suggest authors remove the "freely" from the data availability section, add that an embargo period exists and explain why.

The SEANOE repository was put on hold until the submission progresses in case changes were needed.

The repository manager was contacted and asked to open the dataset for free access to all.

Specific comments:

(lines numbering corresponds to the Author's tracked changes file)

• Line 114: sentence is not clear to me, "Soffer et al. (2020)" is repeated twice. A

suggestion could be: In a recent study, Soffer et al. (2020) compared wave parameters

from the DeepLev's first deployment with simultaneous measurements from a

bottom-mounted ADCP located 48.5 km away at a depth of 26 meters.

Thank you for bringing it to our attention. Correction to the mistake was addressed as suggested.

Change applied in lines 109-110

• line 137: Although I am not a native English speaker, I would suggest this sentence as:

"In practice, the appropriate values for the water properties …"

Thank you, we adopted the suggestion.

Change applied to line 129

• Table 2 caption: I would change the caption from "Summary of quality of data …" to

"Summary of quality indexes of data …" or "Statistics on quality of data …"

Thank you, we adopted the suggestion "quality indexes"

Change applied to Table 2 caption.

TEXT editing:

• There is an inconsistency in figure references that should be fixed. Base on the guidelines (https://www.earth-system-science-data.net/submission.html#figurestables) it should be (Fig.2b) and not (fig 2b), or (Fig. 4) and not (Figure 4), etc.

• the "meter" unit is not used in a consistent way, e.g. line 117: 26 meters, line 125: 12 m (27-39 m). Should be fixed.

Thanks, we went over all the guidelines again and corrected these inconsistencies.

Checked every instance of using units and references to figures.

Thank you for your support and for investing time to help us improve our manuscript.